# Immune Checkpoints in Solid Organ Transplantation

**DOI:** 10.3390/biology12101358

**Published:** 2023-10-23

**Authors:** Arnaud Del Bello, Emmanuel Treiner

**Affiliations:** 1Department of Nephrology, University Hospital of Toulouse, 31400 Toulouse, France; 2Metabolic and Cardiovascular Research Institute (I2MC), Inserm UMR1297, CEDEX 4, 31432 Toulouse, France; 3Faculty of Medicine, University Toulouse III Paul Sabatier, 31062 Toulouse, France; 4Laboratory of Immunology, University Hospital of Toulouse, 31300 Toulouse, France; 5Toulouse Institute for Infectious and Inflammatory Diseases (Infinity), Inserm UMR1291, 31024 Toulouse, France

**Keywords:** immune checkpoints, organ transplantation, tolerance, immunotherapy

## Abstract

**Simple Summary:**

The immune system spontaneously recognizes and destroys foreign cells and organs when grafted into a genetically different individual. Organ transplantation is only successful because of the use of life-long immunosuppressive medications, which comes at the cost of severe toxicities. Thus, a major breakthrough in transplantation would be to be able to educate the immune system to accept grafted organs in the long term. A possible way to do that would be to exploit a physiological retro-control of the immune cells, which is based on the timely and coordinated expression of cell-surface receptors with inhibitory activities. In cancer, blocking these receptors (or Immune Checkpoints) boosts the anti-tumor functions of certain immune cells (the T-lymphocytes), with highly significant clinical benefits. Thus, it is likely that opposite actions, such as increasing the expression or the function of these receptors, would result in the dampening of the immune response against foreign organs. Further, this effect would be specific, sparing the protective immune response against pathogens. In this review, we will provide a summary of the current knowledge on the role of immune checkpoints in the context of organ transplantation and explain why and how these pathways could be manipulated to the benefit of patients.

**Abstract:**

Allogenic graft acceptance is only achieved by life-long immunosuppression, which comes at the cost of significant toxicity. Clinicians face the challenge of adapting the patients’ treatments over long periods to lower the risks associated with these toxicities, permanently leveraging the risk of excessive versus insufficient immunosuppression. A major goal and challenge in the field of solid organ transplantation (SOT) is to attain a state of stable immune tolerance specifically towards the grafted organ. The immune system is equipped with a set of inhibitory co-receptors known as immune checkpoints (ICs), which physiologically regulate numerous effector functions. Insufficient regulation through these ICs can lead to autoimmunity and/or immune-mediated toxicity, while excessive expression of ICs induces stable hypo-responsiveness, especially in T cells, a state sometimes referred to as exhaustion. IC blockade has emerged in the last decade as a powerful therapeutic tool against cancer. The opposite action, i.e., subverting IC for the benefit of establishing a state of specific hypo-responsiveness against auto- or allo-antigens, is still in its infancy. In this review, we will summarize the available literature on the role of ICs in SOT and the relevance of ICs with graft acceptance. We will also discuss the possible influence of current immunosuppressive medications on IC functions.

## 1. Introduction

Immune checkpoints (IC) are cell-surface-expressed molecules regulating leucocyte functions. They are mostly found at the surfaces of lymphocyte subsets, but some of them can be expressed on other leucocyte subtypes or even on non-immune cells. Their inhibitory functions are mostly induced upon interaction with their cognate ligand(s), although some cell-autonomous activities have been described. ICs can have one or more specific ligands, which may themselves be more or less widely expressed, both within and outside the immune system. There are several described ICs, the most widely studied being CTLA-4, PD-1, TIGIT, TIM-3, BTLA, and LAG-3 [1,2,3]. These proteins play physiological regulatory functions through transient activation-induced expression, allowing the fine-tuning of cell activation. However, in chronic settings, such as chronic viral infections or cancer, the sustained expression of multiple ICs is associated with metabolic and epigenetic changes, which favor hypo-responsiveness and antigenic persistence [4,5]. This cell dysfunction, sometimes referred to as “exhaustion”, may be reversed through the blockading of the interaction between ICs and their ligands using monoclonal therapeutic antibodies. Indeed, immune checkpoint blockade (ICB) has emerged in the recent years as a revolutionary approach to anti-tumor therapy and [6], more anecdotally, as anti-infectious immune therapy [7]. Marketed ICBs target CTLA4 (Ipilimumab) or PD-1 (Nivolumab, Pembrolizumab, Cémiplimab) and its ligand PD-L1 (Avélumab, Atézolizumab, Durvalumab). However, multiple ICBs targeting other ICs (or their ligands) are under development, and several are currently being tested in clinical trials for cancers (clinicaltrials.gov).

Beyond cancer and chronic infections, antigen persistence and chronic lymphocyte activation are hallmarks of autoimmune diseases (AIDs). The genetic ablation of ICs in laboratory mice either induces or favors the progression of several AIDs [8,9,10,11,12] whereas in humans, CTLA-4 insufficiency induces a syndromic disease including susceptibility to infections, granulomas, lymphocyte infiltration, and autoimmunity [13]. Further, several genetic variants of ICBs are associated with autoimmunity [14,15,16]. Finally, IC expression is often dysregulated in AIDs, and increased, sustained, IC expression may be associated with a more favorable evolution, at least in specific diseases [17,18].

Another situation of antigen persistence is organ transplantation. This situation, however, greatly differs from those previously mentioned since (1) antigens are introduced into the organism through a single, programmed, chirurgical procedure; (2) antigens are, by definition, exogenous and are not tolerated by the immune system at the time of introduction; and (3) the number of antigen-reactive lymphocytes is huge, exceeding the number of antigen-specific lymphocytes (involved in anti-infectious, anti-tumor, or autoimmunity) by several orders of magnitude. In theory, the persistence of foreign antigens should lead to lymphocyte exhaustion, which could be akin to some sort of tolerance. However, the situation might be more complex, owing in part to the aforementioned specificities of organ transplantation but also to the use of immunosuppressive drugs, which could themselves influence T cell differentiation.

In this review, we will summarize the current and available knowledge on ICs in organ transplantation and highlight challenges and promises of using ICs in the management of transplanted patients.

## 2. The Role of Immune Checkpoints and Their Ligands in Solid Organ Transplantation

### 2.1. CTLA-4

Cytotoxic T-Lymphocyte-Associated protein 4 (CTLA-4, or CD152) was the first described inhibitory molecule for T cells [19]. CTLA4 has gained incredible attention after the demonstration that blocking its interactions with its ligands can be highly beneficial to subsets of patients with cancer, paving the way for the immune therapy of cancer with immune checkpoint inhibitors (ICIs) [20,21]. The administration of such therapies in solid-organ-transplant patients was associated with a dramatic increase risk of acute, severe TCMR [22].

The paramount importance of CTLA-4 in regulating the immune system is demonstrated both by the phenotype of CTLA4 insufficiency (as well as LRBA deficiency, which inhibits CTLA4 expression by disrupting its recycling) in humans [13] and the early deaths of CTLA-4 knock-out (KO) mice [23]. CD80 (B7-1) and CD86 (B7-2) are CTLA-4-binding partners expressed on the surfaces of professional antigen-presenting cells. CTLA-4 functions at least at two separate levels: first, it is transiently expressed in activated T cells and disrupts CD28/B7 interactions, thereby acting as a physiological inhibitor of co-stimulation (Figure 1). In this respect, CTLA-4 is assumed to exert its regulatory effects at the time of T cell priming in lymphoid organs. Second, CTLA-4 is also strongly and constitutively expressed by Foxp3+-regulatory T cells (Tregs), and is crucial to their functions (Figure 2) [24]. This dual action of CTLA-4 is a source of complexity and difficulties to disentangle the cellular mechanisms at play in specific conditions [25]. This roadblock has been partially resolved through the generation of cell-type-specific and conditional KO mice. Indeed, the Treg-specific deletion of CTLA-4 is sufficient for mice to develop a fatal lymphoproliferative disease, showing the crucial role of CTLA-4 for the Treg-mediated regulation of conventional T (Tconv) cells in vivo [26]. On the other hand, two groups developed conditional KO to study the consequences of CTLA deletion at an adult age [27,28]. These studies showed somewhat discordant results, with spontaneous autoimmunity in one model [28] but not in the other [27]. Both sets of data, however, converged on the observation that CTLA-4 expression by T conv, not Tregs, is necessary to control its priming, whereas CTLA-4-deficient Tregs remain fully able to suppress T cell responses; this is made possible through the upregulation of other ICs [27]. Thus, the mechanisms of CTLA-4-mediated inhibition are complex and probably context-specific.

Another difficulty lies in the structure similarity and sharing of binding partners of CTLA-4 with the co-stimulatory CD28 molecule, which makes it difficult to analyze them separately. To overcome the drawbacks of traditional CTLA-4 models, some groups managed to generate CD28-specific antagonists, leaving CTLA-4 alone for analysis. They were then able to show that blocking CD28 significantly delays the acute rejection of kidney and cardiac transplants in baboons and macaques, respectively [29]. More impressively, CD28 antagonist synergized with calcineurin inhibitors (CNI) to increase graft survival over weeks. These effects are associated with Treg infiltration and are thought to be CTLA-4–dependent. Indeed, in a murine model of fully mismatched skin allografts, selective CD28 inhibition is able to inhibit the generation of Donor-Specific Antibodies (DSAs) in a CTLA-4 dependent manner [30]. Moreover, increasing the CTLA-4 half-life at the membrane of activated T cells via chloroquine administration decreases the number of alloreactive T cells and increases the infiltration of Tregs in murine models of heart and skin fully mismatched allografts, resulting in increased graft survival [31]. Thus, CTLA-4 prevents or reduces graft rejection in the absence of CD28 signaling.

There are three common polymorphisms in the CTLA-4 gene, which result in lower CTLA-4 expression and/or the decreased production of a soluble form of CTLA-4, which itself leads to increased T cell activation [32]. These polymorphisms have been studied as possible influencers of acute rejection with mixed conclusions. For example, the CTLA-4 rs3087243*G allele was found to be protective against TCMR in kidney transplant recipients (KTRs) [33]. Other polymorphisms were studied but with no definitive conclusion as to their effect on transplantation [34]. These studies suggest that the consequences of these polymorphisms on CTLA-4 functions may be too tenuous to emerge as independent and significant drivers of transplant rejection. Nonetheless, a recent study performed on a small number of patients identified the association of two CTLA-4 polymorphisms with an increased risk of de novo DSA formation over time [35].

### 2.2. PD-1

Programmed Death-1 (PD-1) is a major inhibitory receptor expressed on subsets of memory cells, activated and exhausted T cells, and Tregs (Figure 2). It can be also expressed on other cell types such as NK cells, B cells, and some myeloid cells. PD-L1 and PD-L2 are PD-1 ligands, the former expressed by multiple cell types and the latter primarily by hematopoietic cells such as DCs. In mature T cells, PD-1 inhibits T cell activation by disrupting both the CD3 and CD28 signaling cascades, thereby controlling proliferation and effector functions (Figure 3). Importantly, PD-1 also controls T cell survival, in part through its effects on metabolism [36]. Several studies showed the importance of PD-1 expression for the development and suppressive functions of induced Tregs [37,38], but the use of a conditional knock-out concluded that PD-1, in fact, negatively regulates the suppressive functions of Tregs [39]. Altogether, the importance of PD-1 in controlling T cell reactivity is well established, and this knowledge has led to the success of PD-1/PD-L1–targeting therapies in cancer treatment [40].

Similar to anti-CTLA-4 antibodies, the use of marketed ICI targeting the PD-1 pathway is associated with an increased risk of acute organ rejection, strongly suggesting that PD-1 physiologically regulates alloreactive T cells in the setting of transplantation [22]. Accordingly, PD-1 was strongly but transiently expressed in kidney-infiltrating T cells in a mouse model of kidney transplantation [41], and its blockade strongly exacerbated tubular injury.

The PD-1 pathway has been the subject of a number of studies aimed at deciphering its role in graft survival and how it could be targeted to dampen rejection. More than 20 years ago, Özkaynak et al. analyzed the effect of PD-1 in a mouse model of cardiac transplantation. Whereas targeting PD-1 with PDL1-Ig fusion proteins alone had no effect on graft rejection, it strongly synergized with IS-targeting T cells, such as Cyclosporin A (CsA) or rapamycin (Rapa), to promote graft survival [42]. Interestingly, PD-1 targeting in co-stimulation-deficient mice (CD28^−/−^) in the absence of IS yielded similar results. Accordingly, PD-1 overexpression on recipient T cells promoted tolerance to fully mismatched cardiac allografts treated with CTLA4-Ig. This effect required PD-L1 expression by donor cells and was donor-specific. The long-term effect was mediated by Tregs [43]. In a model of single allo-antigen skin grafts, with a low frequency of alloreactive transgenic T cells, PD-1 is required for long-term tolerance induced by combined CD28/CD40L blockade [44].

These data suggest that the inhibitory function of PD-1 may not be sufficient by itself to protect against acute rejection; however, it displays a cumulative effect with either T-cell-targeting IS or the absence of co-stimulation and then becomes able to modulate acute and chronic rejection. PD-1 blockade also accelerates heart rejection in fully mismatched animals, but only in the absence of CD28 co-stimulation [45]. Chimera experiments demonstrated the importance of PD-L1 expression on non-hematopoietic donor cells [46] for protection against rejection.

The PD-1/PD-L1 pathway may play a prominent role in controlling allograft vasculopathy. In vitro, smooth muscle cells express PD-L1 upon IFNγ treatment, and its blockade amplifies their proliferation induced by splenocytes. PD-L1 is also expressed in SMC from graft arteries in vivo, and anti-PD-L1 treatment increases the thickening of the intima [47]. Another study reported similar results, using both murine models of single class-I and class-II mismatches. In both models, PD-Ll deficiency on donor non-hematopoietic cells accelerated rejection [48].

In line with this, the artificial expression of PD-L1 by glomerular endothelial cells could delay graft damages upon acute rejection in a rat model of kidney transplantation. This effect was associated with decreased T cell infiltration except for an increased frequency of regulatory T cells [49].

PD-L1 may also be involved in protecting renal tubular epithelial cells (TECs) from alloreactive T cells [50]. TECs express PD-L1 under inflammatory conditions, and this PD-L1 is then able to inhibit alloreactive CD4 and CD8 T cells, at least in vitro. Biopsy analyses show that whereas PD-L1 expression is indeed induced during rejection, it is not sufficient to fully protect against damage. Nonetheless, these observations suggest that increasing PD-L1, and maybe other IC, expression on TECs may be beneficial during kidney transplantation.

PD-1 may be involved in operational tolerance in liver transplant recipients. Plasmacytoid dendritic cells from operationally tolerant liver transplant patients express a high PD-L1/CD86 ratio, as compared with patients under maintenance immunosuppression, which correlates with an increased circulating Treg frequency [51]. This corroborates other works showing the importance of PD-L1 expression on liver DCs to induce spontaneous tolerance to allogenic liver transplantation in mice. Tolerance in these models was dependent on Tregs induced by those PD-L1-positive DCs [52], but also on exhausted PD1 + Tim3+ graft-infiltrating CD8 T cells [53].

In a recent proof of concept, the overexpression of PD-1 together with the inhibitory enzyme Indoleamine 2,3-dioxygenase (IDO) by allograft islet cells promoted survival and the inhibition of acute rejection in a model of C57Bl/6 to diabetic Balb/c transplantation [54]. Inhibition was dependent upon the presence of CD4+ T cells, probably interfering with the alloreactive responses of CD8 T cells and decreased macrophage infiltration. Importantly, similar results could be obtained in larger xeno animal models (pigs into mice or dogs).

Another intriguing finding was reported recently in a mouse model of fully allogenic heart transplantation [55]. In this setting, low-dose treatment with IL-2, combined with the co-stimulation blockading of both the CD28- and CD40L-pathways, prevented chronic rejection induced by transplantation with co-stimulatory blockade alone and increased graft survival. Interestingly, this effect was accompanied by both an increased graft infiltration by Tregs as well as the presence of circulating exosomes carrying the PD-L1 and the CD73 molecules, thereby suggesting immuno-regulatory properties. Although the demonstration that these exosomes are directly involved in preventing rejection is lacking, the role of IC-expressing exosomes in suppressing immune responses against tumor cells has been described [56]. Thus, those findings may be highly relevant and open new avenues of investigations in the settings of allo-immune responses.

### 2.3. Tim-3

Constitutively expressed by dendritic cells and regulatory T cells [57], T-cell-immunoglobulin-and-mucin-containing protein-3 (Tim-3, or CD366) is induced early after activation on conventional T cells [58]. Tim-3 expression by Tregs may play a major role in their suppressive function, at least in specific contexts [59,60]. In contrast to CTLA-4 or PD-1, Tim-3 expression is confined to the Th1 and Th17 cell subsets of CD4+ T cells, and to IFNγ-producing -CD8+ T cells [58,61,62]. These observations suggest a specific role for Tim-3 in the regulation of the more inflammatory subsets of T cells. Another peculiarity of Tim-3 is the existence of several ligands, including galectin-9 (Gal9), Carcinoembryonic-antigen-related cell-adhesion molecule 1(CEACAM-1), Phosphatidylserine (PtdSer), and high-mobility group box 1 (HMGB1), showing wide cellular and tissue expression. However, Gal9 seems to be the major Tim-3 ligand. Tim-3 signaling directly interferes with the CD3 signaling cascade. However, the mechanisms by which Tim-3 regulates T cell functions are poorly understood; the current hypotheses propose that Tim-3 displays a dual stimulatory/inhibitory function depending on the interaction with an intra-cytoplasmic signaling adaptor, Bat3 (Figure 4). Tim-3 is highly expressed on Tregs in cancer, leading to increased suppressive functions, and tumor growth in murine cancer models [63]. Tim-3, together with other ICs such as PD-1 and CTLA-4, is highly expressed on exhausted T cells in viral infections [64,65] and cancer, i.e., cells that become hypo-responsive in the context of chronic stimulation (see below). Hence, it is a likely candidate for future immunotherapies [66].

It was shown 20 years ago, in a seminal paper, that Tim-3 was involved in a model of transplantation tolerance to islet allografts induced by a combination of donor-specific transfusion and CD40L blockade [67]. Similarly, blocking Tim-3 largely accelerated acute rejection in a murine model in the absence of CD28/B7 co-stimulation [57]. Thus, akin to PD-1, the influence of Tim-3 is uncovered when co-stimulation pathways are blocked.

The therapeutic potential of the Tim-3/Gal-9 pathway was suggested in a mouse model of ischemia-reperfusion injury on liver transplantation [68]. The transgenic overexpression of TIM-3 showed a strong hepatoprotective effect, mediated by interaction between Tim-3 expressing recipient CD4 T cells and Gal-9-expressing hepatocytes.

Accordingly, Gal-9 was shown to mediate protective effect in several models, including overexpression in islet transplantation [69], rat liver transplantation [70], skin transplantation [71], cardiac allograft along rapamycin treatment [72], and constitutive expression in corneal allografts [73]. Thus, Tim-3 appears to be a strong candidate for the development of candidate drugs in transplantation.

### 2.4. BTLA

B and T Lymphocyte Attenuator (BTLA) belongs to the CD28 superfamily of receptors and is broadly expressed on T, B, and dendritic cells. In contrast with most inhibitory receptors, BTLA4 is highly expressed on quiescent, naïve T cells, and its expression tends to decrease with activation. BTLA ligand is the Herpes Virus Entry Mediator (HVEM), a member of the TNF-R superfamily expressed on various leucocyte subsets. BTLA regulates cell functions in part through the ITIM domain encoded in its cytoplasmic tail, interfering with T cell signaling (Figure 5) [74].

It was shown as early as 2005 [75] that BTLA has a dominant role in the acceptance of partially mismatched heat allografts in a mouse model as alloreactive cells infiltrating the grafted heart show strong BTLA expression. BTLA expression is decreased in KTR with biopsy-proven acute rejection, and its overexpression was shown to be protective against rejection in a rat model of KT [76]. BTLA acts primarily to dampen alloreactive CD4 T cell activation, although a detailed description of the BTLA pattern of expression in this model is lacking. Graft survival with BTLA overexpression was further sustained with the use of a concurrent blockade of a major co-stimulatory pathway with belatacept [77], highlighting, again, the strong synergy between IC signaling and co-stimulation blockade.

### 2.5. TIGIT

TIGIT (T cell immunoreceptor with Ig and ITIM domains) has recently been implicated in events linked to transplant tolerance and rejection and as potential target for immunotherapy. TIGIT is expressed on subsets of memory T cells, Tregs, and NK and B cells (Figure 2). It binds to three different ligands, CD155, CD112, and CD113, with different affinities. Importantly, TIGIT shares the same ligands with CD226 (DNAM-1), a co-stimulatory receptor expressed on T and NK cells (Figure 6). TIGIT and CD226 thus compete for ligand binding, with TIGIT winning the race owing to its higher affinities for these ligands. TIGIT may regulate T cell functions through multiple mechanisms as its expression directly inhibits T cell signaling, but it also induces IL-10 production by DCs after interacting with their ligands [11,78,79,80,81]. In Tregs, TIGIT expression is associated with an increased suppressive capacity towards Th1 and Th17 pro-inflammatory cells and controls the functional stability of Tregs [82,83] (Figure 2).

Kidney transplant recipients show a gradual accumulation of T cells expressing TIGIT as well as other IC over the first 6 months following transplantation [84]. Accordingly, our own studies found increased TIGIT expression over time (up to 5 years) in stable kidney transplant patients, suggesting a potential role in functional T cell hypo-responsiveness that develops in these patients [85,86]. In contrast, in a cohort of stable KT patients, TIGIT was expressed predominantly in polyfunctional memory T cells [87]. TIGIT-expressing T cells gradually disappeared over time, leading to reduced donor-specific responsiveness. The reasons for the discrepancies between this and the aforementioned studies are unclear, but they all point to a role for TIGIT in controlling T cell responsiveness in the setting of kidney transplantation.

Using a mouse model based on the adoptive transfer of OVA-specific transgenic T cells and skin grafting (minor antigenic mismatch), Hartigan et al. showed that a TIGIT agonist can synergize with a co-stimulation blockade (with a CTLA-Ig) to strongly protect against rejection [88]. This effect is associated with increased Treg intra-graft infiltration, and evidence points to a major role of TIGIT expression by Tregs in this model. However, the same group showed previously that TIGIT was strongly expressed by human belatacept-resistant CD4 and CD8 memory subsets and that TIGIT agonists induced apoptosis of these cells in vitro [89]. Thus, TIGIT signaling might be beneficial to both by enhancing Treg functions and inducing apoptosis signaling in conventional graft-infiltrating T cells.

### 2.6. CD244

CD244 (SLAMF4), also known as 2B4, belongs to the SLAM (Signaling Lymphocytic Activating Molecules) family. It is mostly expressed on NK cells but also on subsets of CD8+ T cells and other leucocyte subsets. CD244 binds CD48, another SLAM, through not only *trans* but also *cis* interactions (Figure 7). CD244 is peculiar in its ability to transmit both stimulatory or inhibitory signals in a process that may be regulated by binding to different adaptor-signaling molecules (SAP or EAT-2) [90,91].

A role for 2B4 was demonstrated in a fully mismatched model of Balb/c to B6 skin grafting [92]. Inhibiting CD28 induced 2B4 expression on alloreactive CD8 T cells, and this was partially responsible for the efficacy of the CD28 blockade. Interestingly, an intact CTLA4 signaling pathway was required for this effect.

### 2.7. Other Inhibitory Receptors

Lymphocyte Activation Gene-3 (LAG-3, or CD223) was the second inhibitory co-receptor discovered, after CTLA-4 [93]. Like many others, it is upregulated upon TCR-mediated activation on T cells and expressed on subsets of Tregs (Figure 2). Similarly to PD-1, initial studies showing the importance of LAG-3 in Treg functions [94] were balanced by more recent work suggesting that it may, in fact, limit their regulatory functions [95]. Structurally related to the CD4 molecule, it binds MHC class II molecules as well (Figure 8), although other ligands have also been recognized (recently reviewed in [96]). LAG-3 has been suggested to control GVHd in allogenic stem cell transplantation [97] and only one recent paper (not peer-reviewed yet) also suggests it may function in organ transplantation. LAG-3 deficiency on either T or B cells results in prolonged graft survival in fully mismatched kidney transplantation through the inhibition of Ab-mediated rejection [98].

B7-H4, a member of the B7 receptor family, was shown to accelerate the rejection of fully mismatched cardiac allografts in co-stimulation-deficient mice [99]. Accordingly, in a mouse model of allogenic pancreatic islet transplantation in diabetic B6 mice, B7-H4 overexpression in islets was protective in long-term survival [100]. It was further suggested that B7-H4 induced donor-specific T cell unresponsiveness, a process partially dependent on regulatory T cells [101].

B7-H5, also known as V-Domain Ig Suppressor of T cell Activation (VISTA), is an inhibitory receptor expressed on leucocytes, including lymphocytes but also, and mostly, myeloid cells [102]. Kidney-resident macrophages constitutively express VISTA, which serves as a checkpoint for T cell proliferation and function upon stimulation in vitro in ischemia reperfusion [103]. VISTA expression by resident macrophages is necessary in vivo for tissue-repair after ischemic injury. This may be highly relevant to kidney transplantation, and VISTA may be a promising new target as an IC in many contexts [104].

## 3. T Cell Exhaustion in Solid Organ Transplantation

Lymphocyte exhaustion has been described, initially, as a state of CD8 T cell hypo-responsiveness developing in the face of chronic viral infection in mice [105]. This observation has been extended to human chronic viral infections and, mostly, cancer, where lymphocyte exhaustion is thought to explain the inefficiency of tumor-infiltrating lymphocytes in actually clearing cancer cells [4]. A wealth of studies have described the epigenetic, metabolic, phenotypic, and functional features associated with exhaustion in various settings [5]. Most studies describe exhaustion in CD8+ T cells, as the result of both antigen persistence and insufficient CD4+ help [106,107], amongst many parameters [4]; CD4+ T cell exhaustion is less well described [108,109].

The most recent theories postulate that exhaustion actually reflects a gradient of differentiation states for T cells, ranging from early (or precursors) to late-exhausted, with a range of dysfunctional features, which may in fact be a way for T cells to adapt to chronic antigen exposure. In other words, when T cells face an antigen they are not able to clear, they engage in a differentiation process, which allow them to persist, control—to some extent—the antigenic insult, and avoid overt immunopathology [110].

Theoretically, the continuous presence of allo-antigens in SOT may induce a state of exhaustion in alloreactive T cells. As a matter of fact, it has long been shown that alloreactive T cells from some transplanted patients evolve into a state of donor-specific hypo-responsiveness over time, which correlates with better graft function [111,112,113]; the mechanisms behind this are not clear [85,114,115]. The appearance of exhaustion in donor-specific alloreactive T cells could be indicative of some level of adaptation of the recipient immune system to a foreign tissue and open the way to new strategies to attain tolerance to transplanted organs.

Ten years ago, an intriguing paper showed that exhaustion in CD4 T cells mediated allograft tolerance in a model of cardiac transplantation, which could be reversed by anti-PD-1 [116] Surprisingly, exhaustion was induced upon impaired leucocyte migration in a selectin-deficient model.

Zhou et al. were able to induce spontaneous T cell exhaustion in two models of skin transplantation [117]. In the first model, based on a male-to-female transplantation, large-sized, but not small-sized, skin grafts were accepted. Male-antigen-specific recipient T cells became exhausted in large-sized transplants only, as revealed by the induction of multiple-IC expression (PD-1, LAG-3, Tim-3, 2B4), coupled with hypo-responsiveness to mitogenic stimulation in vitro. Similar findings were recapitulated in the other model, in which the adoptive transfer of transgenic T cells from B6 mice, specific for the major Balb/c antigen I-Eα, was performed. These transgenic T cells became exhausted, but only after transfer into Balb/C * B6 F1 mice, and not in B6 mice grafted with Balb/c skin. Exhaustion was clearly demonstrated based on phenotype and transcriptome analysis, but also upon failure to reject Balb/c skin grafted onto lymphopenic recipients. Thus, spontaneous exhaustion can be demonstrated in models where (1) alloreactive T cells exist in small numbers with a narrow repertoire and (2) allo-antigens are in very large quantities.

In humans, both PD-1 and TIGIT expression levels increase in mCD4+ T cells in KTR [85,118], being suggestive of T cell differentiation towards exhaustion. More strikingly, a longitudinal study performed over the first 6 months after KT in patients showed an increase in the number of exhausted T cells, which correlated inversely with graft fibrosis and positively with eGFR [84]. In liver transplants, patients demonstrating operational tolerance (i.e., graft survival in the absence of immunosuppressive medication) showed the overexpression of both the TIM-3 ligand Gal9 and PD-L1, which could foster the development of T cell exhaustion [119].

Thus, reports of T cell exhaustion in solid organ transplantation are scarce. Further, the fact that, in most cases, the discontinuation of immunosuppressive treatments leads to organ rejection argues against a state of T cell exhaustion, which should persist after drug removal. This leads to the next question: what are the consequences of immunosuppressive treatments on the regulation of IC expression, and, more generally, on the development of exhaustion?

## 4. Interactions between Immunosuppressive Drugs and IC Expression and Function

It is highly likely that immunosuppressive drugs may have an impact on IC expression and T cell exhaustion. In the mouse model of chronic LCMV infection (a model for antigen-specific exhaustion), a complex interplay between PD-1 and mTOR is demonstrated, whereby rapamycin inhibits the effect of PD-1 blockade [120]. This suggests that mTOR regulates T cell exhaustion in this model. However, blocking mTOR at the beginning of a T cell response increases the response and inhibits exhaustion [121]. The chronic administration of rapamycin to healthy aged mice decreases the expression of PD-1 and LAG-3 and increases proliferative capacities, suggesting the reduction of “natural” exhaustion [122].

Similarly, NFAT promotes T cell exhaustion in CD8 T cells, suggesting that calcineurin inhibitors, which block NFAT translocation, can be harmful to exhaustion induction [123]. Accordingly, it was very recently demonstrated that calcineurin blocks Tox expression in alloreactive T cells after HSCT, thereby inhibiting the terminal differentiation of these T cells to exhaustion [124].

In a study analyzing the effect of long-term treatment with IS after kidney transplantation [125], it was found that there was increased PD-1 expression but decreased PD-L1 and CTLA-4 expression. Importantly, the dynamics of IC modulation differed between patients treated with CNi or mTORi, with higher PD-1 and CTLA-4 levels in the latter subset. Comparable data were obtained in a study focusing on CD57 + PD1-CD45 T cells, showing the upregulation of PD-1 expression upon in vitro stimulation in the presence of rapamycin, while tacrolimus (a CNi) blocked this effect [126]. The picture is complexified by other reports showing an inhibitory effect of mTORi on PD-L1 expression by tumor cells [127]. These results suggest that T cell differentiation in graft transplantation is the result of both chronic stimulation and the use of IS targeting different pathways. It also opens the hypothesis that IS, while blocking harmful responses, may also interfere with the development of a state of tolerance or hypo-responsiveness to graft allo-antigens. Thus, both CNI and mTORi use may counteract the differentiation of T cells towards exhausted, hypo-responsive T cells.

On the other hand, another widely used IS agent, mycophenolate mofetil (MMF), may show an opposite effect. Several co-IRs (PD-1, LAG3, TIGIT…) displayed higher expression on blood Treg cells from stable liver transplant recipients treated with tacrolimus (a CNi) + MMF compared to patients treated with tacrolimus only, suggesting that MMF increases IR expression [128]. Further, Tregs from patients treated with the same regimen were more effective at inhibiting T cell proliferation [129], and this effect was partially abrogated when treated with a combination of anti-TIGIT and anti-PD-1.

Finally, induction therapy could also alter the expression of ICs. In their work, Fribourg and colleagues observed an increase in several clusters of CD4 and CD8 exhausted T cells (mainly represented by T cells that coexpressed PD1 and TIGIT). Interestingly, the authors observed an increased percentage of exhausted T cells in patients that received an induction with a T cell depleting agent (anti-thymocyte globulins) comparing with patients that did not [84].

## 5. Harnessing Inhibitory Pathways in Solid Organ Transplantation

Beyond the potential role of ICs in predicting the outcomes, several molecules harnessing ICs are currently in the pipe. The success of IC blockade in subsets of cancer patients has prompted a major endeavor in both academic and industry groups to develop new strategies to overcome T cell exhaustion. The reverse action may be efficient in organ transplantation. Indeed, targeting co-stimulatory pathways such as CD80/86 with belatacept has demonstrated interesting results with respect to the prevention of acute and chronic allograft rejections. It is noteworthy that most murine studies showing the role of ICs such as PD-1 in regulating chronic organ rejection have been performed alongside the blockading of co-stimulatory pathways; the manipulation of inhibitory pathways in the presence of intact co-stimulatory signaling has often yielded disappointing results. This observation may simply reflect the fact that blocking co-stimulation while enhancing inhibition is probably more efficient for the regulation of T cell activation as compared with targeting only one of these pathways. However, it is important to recall that insufficient help from CD4+ is one of the processes leading to CD8+ T cell exhaustion [106,130]. Thus, combining IC activation with co-stimulation blockade may be a synergistic approach because the latter would induce or contribute to differentiating T cells towards a hypo-responsive, exhausted state.

Targeting IRF4 (Interferon Regulatory Factor 4), a transcriptional factor essential to T cell differentiation, promoted transplant tolerance in a mouse model of fully mismatched heat transplantation [131]. IRF4 represses PD-1 expression, and IRF-4 -deficient T cells up-regulated PD-1. Tolerance was initially dependent upon PD-1 since anti-PD1 treatment reversed T cell dysfunction, which progressed to an irreversible state reminiscent of fully exhausted T cells. Thus, targeting this transcription factor may represent a therapeutic strategy, which seems to be achievable in clinics [132].

Recently, a research group showed the ability to manipulate known IC pathways. They engineered nano-vesicles (NVs) that simultaneously express PD-L1 and CTLA-4. After a single intravenous injection, these cargos were found in multiple organs and significantly delayed acute rejection in the absence of IS treatment in both skin- and heart-allograft mouse models [133]. The same group alternatively produced PD-L1-expressing NVs encapsulating low doses of rapamycin as another means of controlling rejection [134].

Another interesting work studied the effect of the transgenic expression of PD-L1 by porcine neonatal islets on the outcome of xenotransplantation in a humanized mouse model [135]. Compared to wild-type tissue, islets overexpressing PD-L1 demonstrated reduced rejection and superior engraftment, showing the potential of manipulating this pathway in xenotransplantation, a promising strategy to overcome the shortage of available human organs [136].

The recent interest in TIGIT has also been highlighted by a study showing that TIGIT-Fc recombinant protein induce sustained skin graft survival in mice via signaling through the poliovirus receptor on macrophages and inducing M2 polarization [137].

Thus, various strategies may be developed to exploit IC pathways, but they will probably require combination with appropriate therapies targeting other pathways. Although it is not yet conceivable to fully avoid the use of TORi and/or CNI, it may be necessary to test, in murine models, the possibility of sparing those agents at specific timings post transplantation.

## 6. Concluding Remarks

The interest in the development of therapies activating IC receptors is growing, particularly in the field of autoimmunity [138,139,140]. There is little doubt that manipulating pathways in the field of organ transplantation is a promising strategy for the prevention of graft rejection, at least in an acute setting. However, a lot of work remains to be performed in order to define which receptors should be targeted and to position these therapies among existing IS protocols according to the types of transplantation and the immunological statuses of the recipients.

As for T cell exhaustion, there is little evidence to date that this process really occurs in transplanted patients. As discussed, it is plausible that IS treatments in fact hamper this development. However, pushing allogenic T cell differentiation along the exhaustion pathway appears to be a promising avenue of research to reach the holy grail for transplantation immunologists, i.e., long-term, specific transplantation tolerance towards the grafted organs.

## Figures and Tables

**Figure 1 biology-12-01358-f001:**
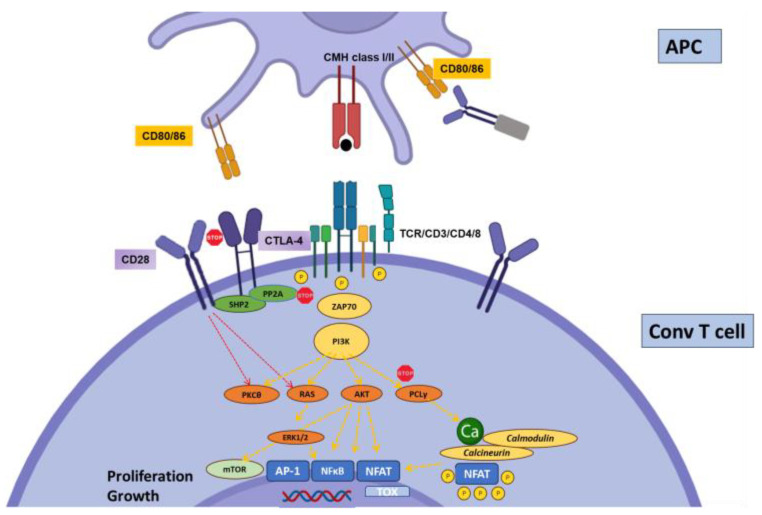
Regulation of conventional T cell activation by CTLA-4. Priming of naïve T cells requires CD28 co-stimulation, which engages multiple stimulatory pathways: PI3K-AKT, PKCθ, and Ras. CTLA4 binds with higher affinity to CD28 ligands CD80 and CD86, thereby disrupting CD28 signaling. Further, the phosphatases SHP2 (SH2 domain-containing tyrosine phosphatase 2) and PP2A (serine/threonine protein phosphatase 2A), bound to the intracellular tail of CTLA4, directly inhibit proximal CD3 signaling.

**Figure 2 biology-12-01358-f002:**
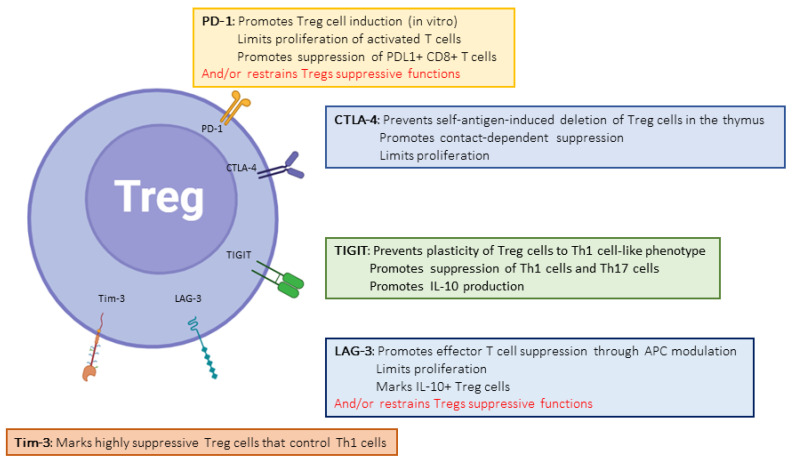
Role of IC expression on regulatory T cells.

**Figure 3 biology-12-01358-f003:**
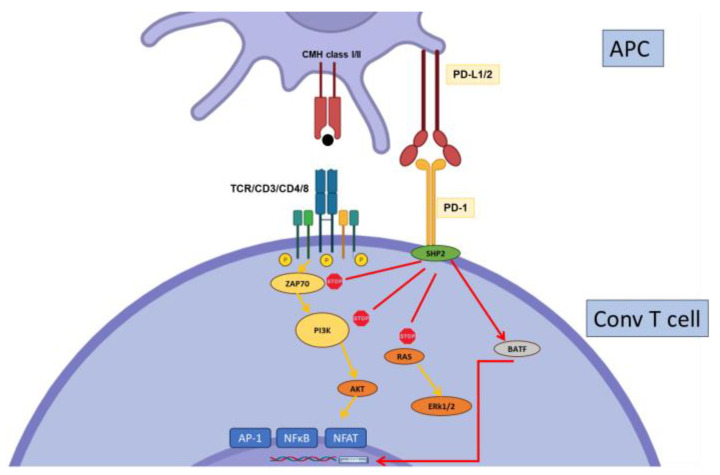
Regulation of conventional T cell activation by PD-1. PD-1 recruits the phosphatase SHP2 to inhibit the CD3 signaling cascade at the level of ZAP-70 phosphorylation and PI3K activation. It also inhibits Ras. SHP-2 can also recruit BATF to inhibit, directly, effector gene transcription. PD-1 may also interfere with CD28 signaling.

**Figure 4 biology-12-01358-f004:**
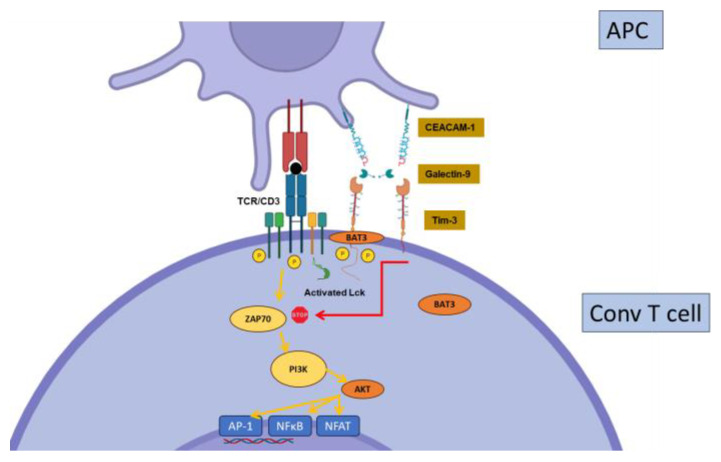
Regulation of conventional T cell activation by Tim-3. Tim- interacts with HLA-B-associated transcripts 3 (BAT3), which maintain lck in an activated form and promote CD3 signaling. In the presence of Gal-9, BAT3 is released from Tim-3, resulting in inhibition of CD3 signaling via an unidentified mechanism.

**Figure 5 biology-12-01358-f005:**
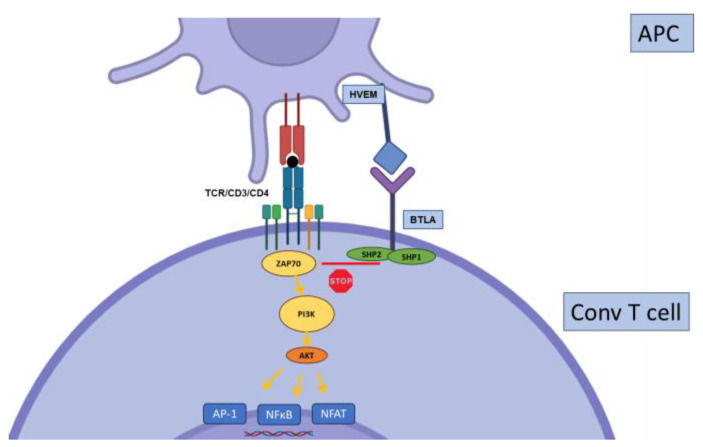
Regulation of conventional T cell activation by BTLA. BTLA interacts with HVEM at the surface of APC. Once activated, it recruits the phosphatases SHP1 and SHP2 to inhibit the proximal CD3 signaling. Further, HVEM is also able to transduce inhibitory signals to the APC (not depicted).

**Figure 6 biology-12-01358-f006:**
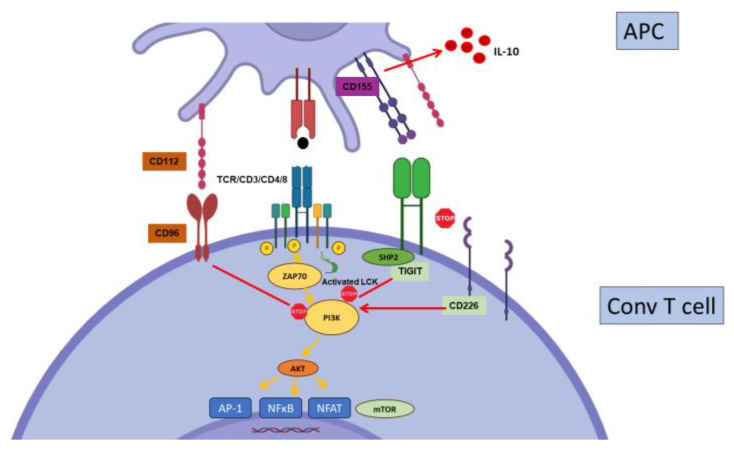
Regulation of conventional T cell activation by TIGIT. CD226 is a major co-stimulatory molecule for T cells. TIGIT binds to the same ligand (CD155, CD112, or CD113) with higher affinity, thereby disrupting the CD226 signaling. TIGIT also mediates direct inhibitory signaling through the recruitment of SHP1/SHP2. Finally, reverse signaling through TIGIT ligands may induce IL-10 production by DCs, making them tolerogenic.

**Figure 7 biology-12-01358-f007:**
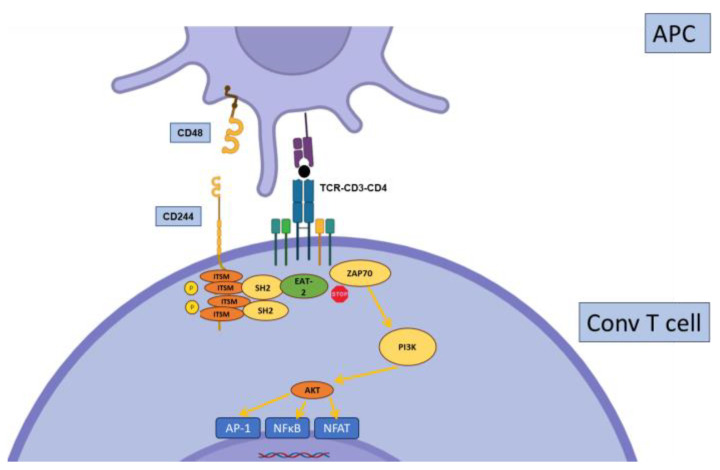
Regulation of conventional T cell activation by CD244. Upon binding to its ligand CD48 (either through *trans*- or *cis*-interactions), CD244 recruits the adaptor protein EAT-2, resulting in inhibition of proximal CD3 signaling.

**Figure 8 biology-12-01358-f008:**
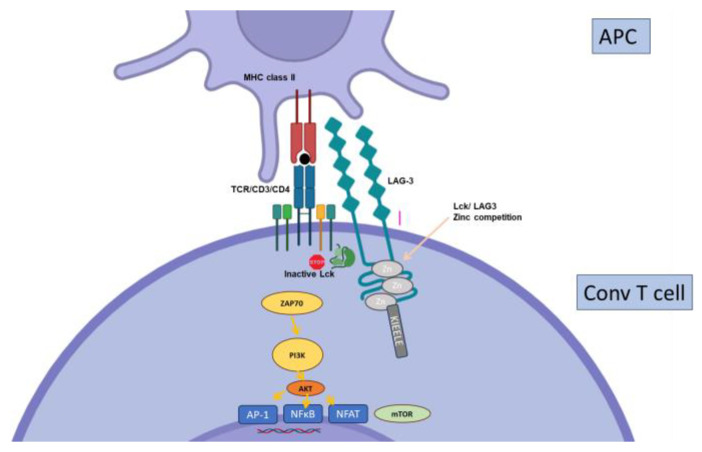
Regulation of conventional T cell activation by LAG-3. The putative mechanisms of action of LAG-3 primarily involve a repetitive C-terminal EP motif, responsible for disrupting co-receptor–lck interactions, in part through Zn^2+^ sequestration.

## Data Availability

Not applicable.

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
