# Peer review of "Immune Checkpoints in Solid Organ Transplantation"

_biology, 2023, doi:10.3390/biology12101358_

Round 1
Reviewer 1 Report
well written and goed described manuscript on immune checkpoints in solid organ transplantation. Findings are valuable for further studies. Well done.
tha authors can add the study below as a reference to increase the value of manuscript.
Yuksel Y, Yuksel D, Yucetin L, Erbis H, Sarier M, et al. Use of Tacrolimus During PregnancyAfter Kidney Transplantation. Transplant Proc. 2019;51(7):2361-2366. doi:10.1016/j.transproceed.2019.01.150
Author Response
We thank the reviewer for her/his positive and supportive comments. We read with considerable interest the article that was suggested. However, we think it does not fit as a reference in our manuscript, and decided not to include it.
Reviewer 2 Report
The authors present a review of Immune checkpoints in the context of Solid Organ Transplantation.
1) Although I agree that drawing conclusions form CTLA-4 KO mice is difficult considering their phenotype, the generation of conditional CTLA-KO mice has helped to understand the effects of CTLA-4 absence in adult mice as well as in specific cell types. I ask the authors to please add more information about the findings of CTLA-4 in adult murine models and different cell types and their contribution to understanding their role is IC inhibitors.
2) I ask the authors to include more information about PD-L1 (in a separate section or expand the current one), since PDL-1 is also used as a checkpoint inhibitor.
3) In the Tim-3 section please expand on the role of CD4 and CD8 T as well as regulatory T cells.
4) The BTLA/TIGIT/LAG3 sections are very scarce, either they should be included in the "Other inhibitory receptors" or expanded. Also, are there any reports of immune checkpoint inhibitors in B-cells and organ transplant? Recently, a paper studying the role of TIM-1 in B cells was published in nature, in a tumor-model context. Is there something similar in organ transplantation?
5) In the "T cell exhaustion in solid organ transplantation" I ask the authors to expand on the idea of hypo responsiveness (transient) versus exhaustion (permanent). Also, please redact the last sentence to give a better idea of how exhaustion could improve organ transplantation before proceeding to the next section.
6) In the section of "Interactions between immunosuppressive drugs and IC expression and function" I ask the authors to explore the balance between IC and inflammatory cytokines, and how this affects the organ transplantation. e.g., if we treat with a-PD1, which cytokines are increased and how does this reflect on organ transplantation?
7) In the section "Harnessing inhibitory pathways in solid organ transplantation" I ask the authors to explore the findings that have been done in CAR T-cell therapy in the field of organ transplantation as they can provide a plausible treatment in the future.
8) Is T cell exhaustion induction the best way? Is it better to create a state of transient decreased responses? Please discuss about the role of antigen specificity in the context of IC (see TCR signaling inhibition). This can relate to the use of CAR T-cells.
The authors have an interesting review that requires work. With the addition of the aforementioned topics, it will be suitable for publication.
Additionally, the authors need to increase the font size in figure 1 to make it readable. Also, the authors should include a more thorough explanation of the role of Tregs, referenced in figure 2 as it is incomplete in the text.
Author Response
First, we wish to thank the reviewer for her/his thorough reading of our manuscript, and for her/his valuable comments and suggestions to improve it. We have tried to address most of these comments and to add informations/references when requested. For various reasons, we did not concur with every comments however, and did not expand on every topic that was suggested by the reviewer. Altogether, we believe that our manuscript has been significantly improved, thanks to all those suggestions. You will find below a point-by-point and detailed reply to the various revisions requested:
- We agree with the reviewer, and added significant informations on various CTLA-4 deficient models: page 3, lines 104-118.
- The reviewer’s expectations on PD-L1 are not totally clear to us. We focused on PD-1 as a T cell-expressed IC, though we obviously mentioned the role of its ligands, highlighting the importance of PD-L1 over PD-L2 in the context of SOT. PDL1 is indeed targeted by IC inhibitors in cancer patients, but with similar indications than anti-PD1. There are indeed reports that there might be relevant differences between those treatments, but which are barely considered at the moment. However, we took advantage of the reviewer’s comments to go through the literature again and we expanded the PD-1 section with what we believe relevant informations. We hope this will respond to the initial comments. See for page 5 lines 174-177, page 6 lines 211-217, 222-236, page 7 lines 243-253.
- Thank you for the suggestion. We expanded the section: page 7 lines 257-261, lines 265-272
- We agree that the sections devoted to those receptors are reduced as compared to CTLA4 and PD1. However, we wished to give them separate sections there are several reports on their role in transplantation, and we believe they are deserve to be presented separately. Thus, we chose to stick with the overall structure of the manuscript. As for the questions on IC expressed by other populations such as B cells, it is an interesting subject but with very few data available. Further, our manuscript is focused on T cells, and we did not want to expand to much on other populations (this could the topic of a separate manuscript).
- We expanded this section, mostly refering to several previous reports on T cell dysfunction (“hypo-responsiveness) which gradually appear in stable transplanted patients. See page 9-10, lines 390-397.
- The reviewer’s expectations on this section is not totally clear to us. Our aim was to document how the immunosuppressive treatments widely used to prevent graft rejection in patients may or may not impact on IC expression and T cell exhaustion. We believe this section does not need further development on related or unrelated subjects such as the role of cytokines.
- CAR-T cells therapy and its potential use in SOT is a fascinating topic; however, we believe it is out of the scope of this manuscript.
- The reviewer’s expectations are not totally clear to us. The question of TCR specificity is also a fascinating topic but out of the scope of this manuscript.
We have also developed in the various sections the role of IC on Tregs, when needed. We believe this improves the manuscript, as suggested by the reviewer.
As for the figure 1, we have increased the font size to make it more readable. We will also ask the editorial team to see how we can increase the overall size of the figure, and if needed, to split it in 2.

Reviewer 3 Report
The authors provide a comprehensive review of the potential role of immune checkpoints in organ transplantation. They discuss the mechanisms of action of individual immune checkpoints (IC) and their corresponding ligands. Moreover, they elucidate the phenomenon of T cell exhaustion in transplantation, the interactions between immunosuppressive drugs and ICs, and the prospects of harnessing IC pathways in transplantation.
Comments:
This review article is commendable and offers up-to-date insights into the role of ICs in organ transplantation. I have a few minor comments:
1.) The authors have noted that reports of T cell exhaustion in organ transplantation are rare. Zou et al. (reference 79) employed a tetramer system to detect exhausted CD8+ T cells in a male-to-female skin transplant model. In contrast, in the Balb/c to B6 transplant model, the adoptively transferred alloantigen-specific CD4+ T cells (TEa transgenic) were not exhausted. Zou et al., therefore, chose to adoptively transfer the TEa cells into CB6F1 (Balb/c x B6) mice, which led to the exhaustion of the TEa cells. Subsequently, these exhausted TEa cells demonstrated a reduced ability to reject Balb/c skins when re-transferred into lymphogenic mice. To avoid any potential misinterpretation by the readers, it might be beneficial for the authors to detail these experimental conditions. Notably, in the absence of specific treatments and manipulations, T cell exhaustion does not naturally manifest in the Balb/c to B6 skin or heart transplantation models.
2.) Specific treatments and manipulations are required to facilitate the acceptance of MHC fully mismatched skin or heart allografts. Under such circumstances, the term “T cell dysfunction” is often preferred over “exhaustion” to characterize the state of T cells. The application of immune checkpoint blockade can help ascertain if this dysfunctional state of the T cell is reversible or has transitioned to an irreversible stage (Wu J et al., Immunity, 2017, 47:1114-1128; Zhang H et al., Am J Transplant, 2019, 19:884-893).
3.) I would recommend increasing the font size in Figure 1 to enhance readability.
Author Response
We thank the reviewer for her/his positive comments. We thank her/him for her/his suggestions and provide here a point-by-point reply:
- We have now clarified in details the model used by Zou et al., as requested. See page 10 lines 402-414
- We have introduced the work by Shang et al. in the last section of the manuscript. Page 11 lines 489-495
- We have increased the font size as well as the overall size of the figure. If necessary we will ask the editorial team to find a way to make the figure more readable.

Reviewer 4 Report
The article presented provides a very useful summary of immune checkpoints in SOT. Additionally, it provides good evidence as to the importance of each IC type and IS drug on T cell exhaustion in transplantation. The authors also take care to draw conclusions from work both from human and animal studies.
minor revision: Extent the T-cell exhaustion in SOT section with CD8 T cell exhaustion.
The authors well reviewed the IC in SOT and discussed their portential application in this field. Extented and detailed the publiched research on IC in SOT is helpful.
Author Response
We thank the reviewer for her/his positive comments. We have extended the section on CD8 T cell exhaustion, as well as on hypo-responsiveness of T cells in transplanted patients in the section. See pages 9 and 10.

Round 2
Reviewer 2 Report
The authors have addressed some of the comments from my previous review. I believe the manuscript can benefit from the addition of a few more explanations.
1) CTLA-4 part was fully addressed
2) I am satisfied with the additions to the PDL-1 section.
3) In the Tim-3 section please include the reference about the expression of TIM-3 only in the mentioned Th subsets and please make sure to add that Tim-3 is expressed in Tregs and seems to play an important role in the phenotype and functionality of these cells.
4) I am satisfied with the response about B cells in IC during transplantation.
5) The Hypo-responsiveness question was addressed.
6) As the authors mention, usage of checkpoint inhibition is of great importance. However, usage of antibodies blocking cytokines is also of great relevance for treating autoimmune/inflammatory diseases. Are these used in the context of transplantation? Is there any evidence that combining or modulating cytokines through the usage of antibodies can be of benefit for transplantation when in combination with IC inhibitors? I hope this expands on the idea of previous question number 6.
7) Although, I understand that CAR T-cells are a very interesting but apparently unrelated topic. The usage of modified CAR T-cells including PD-1 domains has been tested in cancer context. Are there any of these treatments available for transplantation? Have they been proposed?
8) I understand that antigen specificity and IC therapy is a complex subject and can be further explored in a different manuscript.
Finally, please make sure that figures are of good quality, like you mentioned in your reply.
Author Response
We thank the reviewer for acknowledging our changes to improve the manuscript, as requested. As for the remaining remarks, she/he will find below a point-by-point reply.
3) In the Tim-3 section please include the reference about the expression of TIM-3 only in the mentioned Th subsets and please make sure to add that Tim-3 is expressed in Tregs and seems to play an important role in the phenotype and functionality of these cells.
We have now added some precisions in that section, as requested. See page 6 lines 252-257
6) As the authors mention, usage of checkpoint inhibition is of great importance. However, usage of antibodies blocking cytokines is also of great relevance for treating autoimmune/inflammatory diseases. Are these used in the context of transplantation? Is there any evidence that combining or modulating cytokines through the usage of antibodies can be of benefit for transplantation when in combination with IC inhibitors? I hope this expands on the idea of previous question number 6.
In contrast to the field of autoimmunity and autoinflammation, where inflammatory cytokines are major targets of innovative therapies such as therapeutic monoclonal antibodies and JAK inhibitors, the situation is quite different in SOT.IL-6 is the main focus since its role in organ transplantation is quite well established. Several phase ½ clinical trials have been performed with IL-6 targeting therapies such as tocilizumab and Clazakizumab, mostly as a desensitization agent in highly sensitized patients failing standart therapy protocols. While these trials showed some benefits, only phase 3 trials will yield some kind of definitive answer with regard to the efficacy of such treatments. Thus, we prefer not to speculate on possible combination of anti-cytokine therapies and IC modulation in SOT, as data are clearly lacking.
7) Although, I understand that CAR T-cells are a very interesting but apparently unrelated topic. The usage of modified CAR T-cells including PD-1 domains has been tested in cancer context. Are there any of these treatments available for transplantation? Have they been proposed?
In the context of SOT, CAR T-cell therapy is considered as a way to achieve immunological tolerance through Treg-based therapy. For this purpose, Tregs, usually isolated and amplified from the patient’s blood, are transduced with a CAR construct specific for allo-antigens. First proposed in the context of GVHd, preclinical studies have been performed in the context of SOT, with interesting results, paving the way for a phase ½ clinical trial in humans (reviewed in Gille et al., Frontiers in Immunology 2022; see also Lamarche et al., Kidney Int Rep 2022). This CAR Treg therapy is promising but still in its infancy, with numerous questions and challenges ahead: source of Tregs, phenotypic and functional stability of CAR Tregs, identification of targeted allo-antigens, …In this regard, the questions of IC expression by these CAR-T regs, how they interfere in the process and do we need to regulate their expression are (or will be) relevant but it seems very premature to address these questions, at least in the context of this review.
Finally, please make sure that figures are of good quality, like you mentioned in your reply.
Acknowledging the remark, we have now separated each figure to increase their size and make them easily readable.

Round 3
Reviewer 2 Report
I agree with the authors responses to my comments.
The manuscript is suitable for publication with minor corrections.
Please make sure the TIM-3 references are in the correct order, especially reference 58 can be cited at the end of the newly added yellow paragraph.
Figure 6 could not be displayed in the new version of the manuscript. Please ensure that it is correctly formatted in the final version.
Author Response
Dear reviewer,
we have now replaced the references for Tim3 in correct order.
We apologize for the mistake in inserting figure 6- we have now inserted the correct figure
We thank the reviewer for her/his help in editing this manuscript.
